# Maximum Efficiency Despite Lowest Specific Speed—Simulation and Optimisation of a Side Channel Pump [†]

**Markus Mosshammer [1],*, Helmut Benigni [1],*, Helmut Jaberg [1],* and Juergen Konrad [2]**

[1]   Institute of Hydraulic Fluidmachinery, Faculty of Mechanical Engineering, Graz University of Technology. Kopernikusgasse 24/IV, 8010 Graz, Austria

[2]   Dickow Pumpen GmbH & Co. KG, Siemensstraße 22, 84478 Waldkraiburg, Germany; konrad@dickow.de

*   Correspondence: mosshammer@tugraz.at (M.M.); helmut.benigni@tugraz.at (H.B.); helmut.jaberg@tugraz.at (H.J.); Tel.: +43-316-873-8074 (M.M.)

[†]   This paper is an extended version of our paper published in the Proceedings of the Conference on Modelling Fluid Flow (CMFF) 2018, Paper No. 31.

**Abstract:** Side channel pumps provide high pressure at relatively low flow rates. This comes along with a quite low specific speed and thus with the known disadvantage of a quite poor maximum efficiency. This paper describes the detailed analysis and optimisation of a typical 1-stage side channel pump with an additional radial suction impeller by means of computational fluid dynamics (CFD) simulations. In a first step, the model was successively generated and it was obvious that it has to contain all details including suction impeller and main stage (both 360° models) as well as the pressure housing and all narrow gaps to provide useful simulation results. Numerical simulations were carried out in a stationary and transient way with scale resolving turbulence models to analyse the components in detail. Finally the CFD-simulations were validated with model tests. For the optimisation process it was necessary to generate a reduced numerical model to analyse the effects of more than 300 geometry variations. The findings were then combined to establish the desired objectives. Finally the best combinations were validated again with the full numerical model. Those simulations predict a relative efficiency increase at best efficiency point (BEP) and part load >30% with respect to all given limitations like identical head curve, suction behavior, and dimensions.

**Keywords:** CFD; low specific speed; optimisation; side channel pump; simulation; (U)RANS

## 1. Introduction

Providing high pressure at relatively low flow rates, side channel pumps are often used when choosing between a displacement pump and a centrifugal pump like Grabow [1,2] has shown in Figure 1. In this paper, the efficiency is plotted versus specific speed $n_q$ (defined according to Equation (1) instead of the specific speed $\sigma$ (defined according to Equation (2)) used by Grabow as the specific speed $n_q$ is more common in the pump industry. For side channel pumps, the specific speed is approximately between $n_q$ = 1 [rpm, m³/s, m] and $n_q$ = 10 [rpm, m³/s, m] as marked in Figure 1.

$$n_q = n \cdot \frac{Q^{1/2}}{H^{3/4}} \tag{1}$$

$$\sigma = \frac{\phi^{1/2}}{\psi^{3/4}} \approx \frac{n_q}{157} \tag{2}$$

$$\phi = \frac{\frac{Q}{A_{SC}}}{r_a \cdot \omega} \tag{3}$$

$$\psi = \frac{2 \cdot \Delta p}{\rho \cdot (r_a \cdot \omega)^2} \tag{4}$$

Beside the main advantages of an outstanding suction performance and the capability of pumping fluids with high gas loads, the drawback of side channel and peripheral pumps is a quite poor hydraulic efficiency of <50% as shown in Figure 1. It is remarkable that both for large specific speeds and small specific speeds a very good efficiency can be reached with centrifugal and displacement pumps, whereas in-between no machine type exists reaching a very good efficiency.

In the past, when efficiency was often negligible, this fact was simply accepted. Nowadays, companies are pushed by law to increase the efficiency of their products, e.g., by the Energy Efficiency Directive of the European Union [3] or the Paris Agreement. As pumps almost require 20% of the world electric power consumption [4], they offer a huge potential for saving energy. Though many manufacturers have recognized this trend and improved their main models, some exotics like side channel pumps were neglected in the past.

Due to the fact that side channel pumps neither belong to displacement pumps nor to centrifugal pumps, the technological progress in both fields is not applicable. Actually, the design of most side channel pumps available on the market today is quite old (>30 years) as are the few available design guidelines. Also modern design techniques like numerical simulations have therefore not been applied often. Even in cases when simulation results were experimentally validated [5,6], the deviations in head and efficiency are often not satisfying.

This paper describes the detailed analysis of a typical 1-stage side channel pump with an additional radial suction impeller by means of computational fluid dynamics (CFD) simulations and shows the required numerical effort to generate reliable results. For that purpose it is necessary to analyse the occurring losses between inlet and outlet to understand the loss mechanisms in all parts. Therefore it was necessary to increase the numerical model successively—ending with a quite complex model to get an accurate loss analysis for each part.

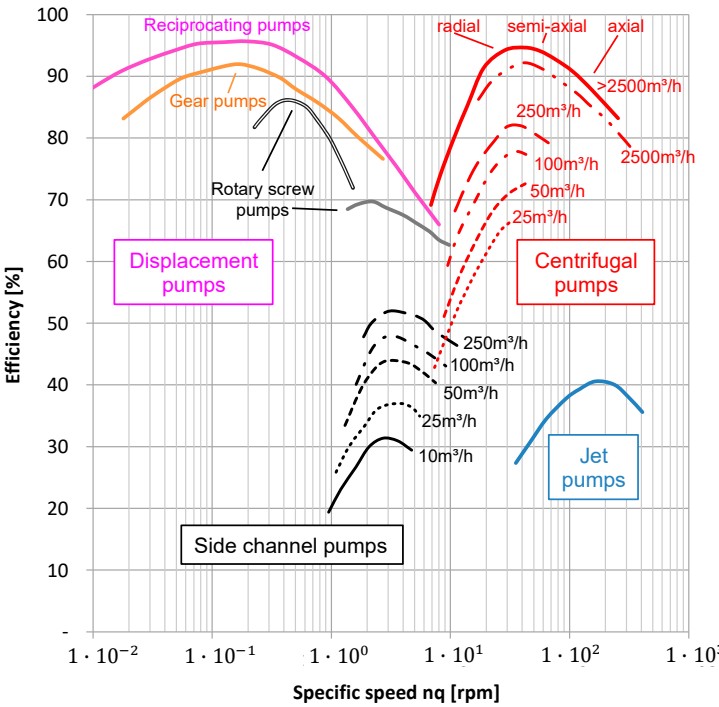

**Figure 1.** Maximum efficiency depending on specific speed and flowrate [2].

## 2. Theoretical Flow in a Side Channel Pump

Side channel pumps are rather to be regarded rotating displacement pumps according to their functioning although they can easily be mistaken as centrifugal pumps because of the design principle. Their functioning is related to a multistage radial pump as the liquid is entrained in the circumferential direction by the rotation of the impeller and is sucked in through the suction opening (Figure 2). Inside the impeller (in the rotating system) energy is applied to the medium by this acceleration and at the same time the fluid moves outwards as it undergoes a centrifugal force and moves into the side channel (stationary system). In this stationary system the medium is decelerated and moves towards the machinery axis. In the stationary system, the kinetic energy is changed into pressure by deceleration. Closer to the impeller center the flow is directed backwards into the impeller, is again dragged away and kinetic energy is anew transferred into the liquid, which moves outward and back into the side channel, where again the kinetic energy is changed into pressure, and so on. This is the so called "internal multistage" effect which is the main characteristic of the side channel pump.

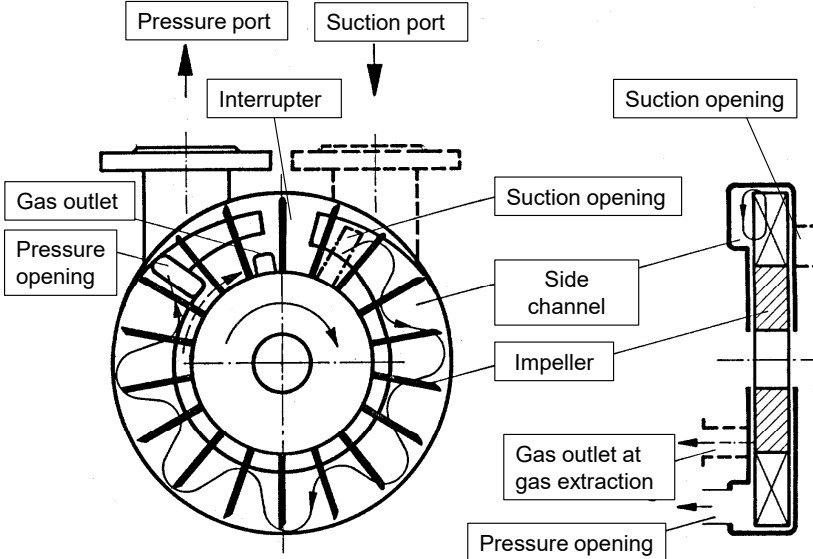

**Figure 2.** Transport principle of a side channel pump.

In the meridional view the fluid moves in a circle (see Figure 2 right) and in the cross sectional view (Figure 2 left) in a form of wiggly line until it leaves through the pressure side which is separated from the suction side by the so called interrupter. This multiple addition of energy in the impeller channels and the transfer of kinetic energy into pressure in the side channels allow the side channel pump to reach large pressure heads for very low flow rates, and its characteristic is comparatively steep.

The principal design of a side channel pump therefore consists of

- Impeller (commonly open design)
- Side channel
- Interrupter (separating low and high pressure side)
- Suction and pressure port (axial/axial, axial/radial, radial/axial, radial/radial)

Until now, two theories, "theory of flow filament" and "theory of shear stress" are still used to describe the flow inside a side channel pump:

(a)　The theory of shear stress, also called theory of turbulence and/or mixing theory is based on the induced shear stress of the impeller on the fluid in the side channel—the fluid is dragged against the existing pressure gradient. Whether the centrifugal forces are included (Pfleiderer [7], Senoo [8–10]) or not (Iverson [11]), the main functional principle is based on displacement. Fluid

with high velocity exits the impeller to the side channel and is there decelerated. In parallel, fluid enters the impeller from the side channel and gets accelerated. This mixing procedure of high turbulence fluid from the impeller and low turbulence fluid from the side channel is repeated multiple times along the side channel and the physical background of this theory.

(b)   The theory of flow filament is based on the helical flow between side channel and impeller. The reason behind the helical flow is based on one hand on the different velocities of the fluid in the impeller and the side channel and on the other hand on the difference between the centrifugal forces of the rotating and the stationary parts. It is assumed, that the impeller induces an angular momentum racua (in the rotating system) which returns near the hub with an angular momentum of ricui (in the stationary system). The difference of the induced angular momentum on the participating fluid volume of the side channel ASC is the theoretical head of of a single "stage" of the side channel pump.

Though they are based on the different physical principles, "common design guidelines" for side channel pumps like from Grabow [12] or Surek [13] use both theories. The most influential dimensions of both theories are the side channel area ASC and the diameter/radius of the impeller. The main dimensions of a side channel pump are described in Section 5.1. A good overview of geometrical influence on side channel pumps is given in [14].

## 3. Methodology of Numerical Simulations

In this chapter the methodology used to simulate a side channel pump is described, starting with the generation of a 3D-model, meshing, and preparation of a suitable numerical setup up to the used post-processing.

### 3.1. Preparation of the Geometry and Mesh Generation

In a first step, the real pump had to be transferred in a 3D-computer-aided design (CAD) model. For that, drawings and dimensions of the pump were used in addition to a 3D-Laserscan of the impeller. Based on the 3D-CAD model, the geometry was prepared for meshing. As an example, the impeller is mentioned as it consists of 24 identical blades (1 blade equals to a section of 15°), where only one was used for meshing. To generate a suitable, structured mesh, it was necessary to split the geometry (we call it blocking) as shown in Figure 3 on top right. Those blocks were meshed separately in ANSYS Meshing (Canonsburg, PA, USA), but each of them with a conformal interface to maximize the resulting mesh quality and to overcome the need of an arbitrary interface. The same procedure was used at the suction impeller, where 1 out of 6 blades was used to generate a structured mesh in ANSYS TURBOGrid.

Besides a correct modelling of the impellers, it is vital to mention, that the resulting head and efficiency of a pump, and especially of a side channel pump, relies heavily on the existing gaps. In Figure 4 the radial gap of 0.3 mm at the suction impeller (left) and the even more narrow axial gap of 0.1 mm at the impeller (right) are shown. To get an impression of the size of the gaps, it is vital to mention that they equal 0.25%, respectively 0.08% of the impeller diameter. Those gaps in addition to the hub and shroud cavities and the side channel itself are of essential importance for reliable simulation results.

The full CFD-model of the side channel pump as shown in Figure 5, including amongst others all narrow gaps and 360° models of suction runner and impeller, consists of 15 mio. nodes. As already mentioned, mainly structured grids for the impellers, side channel, suction piece, and gaps were used.

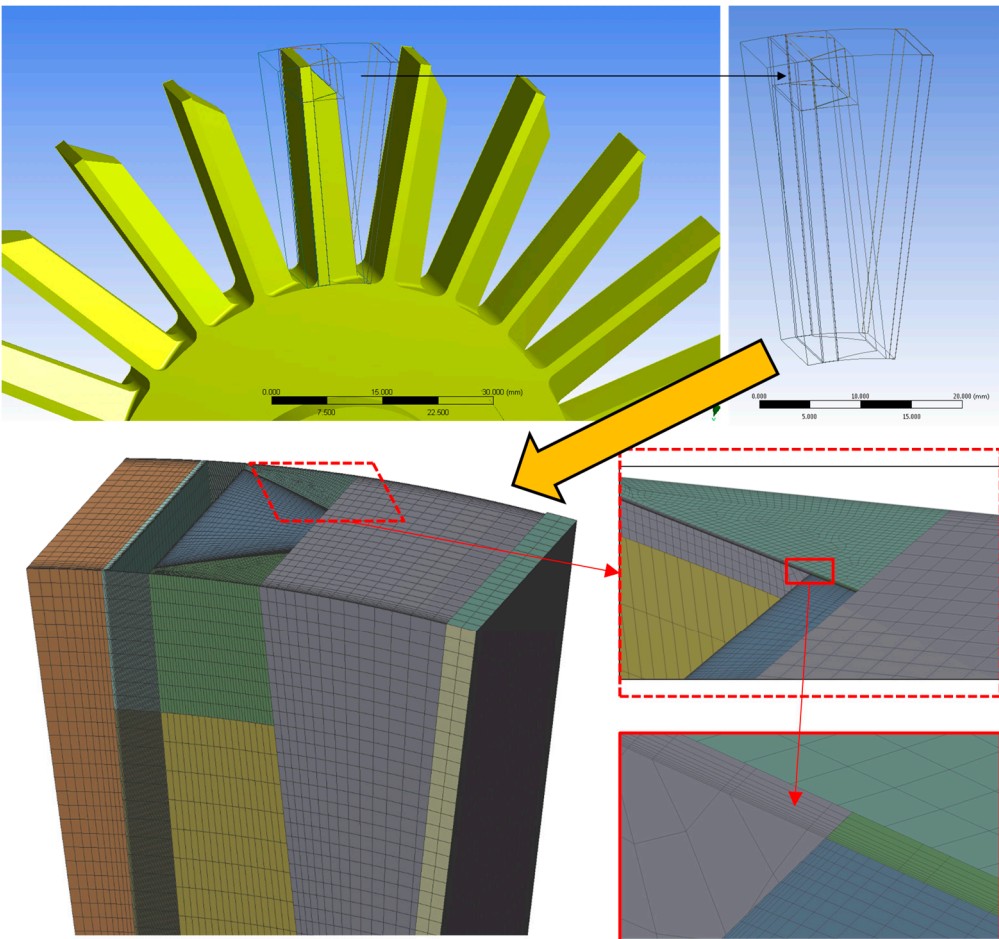

**Figure 3.** Preparation of the geometry and mesh generation of the impeller.

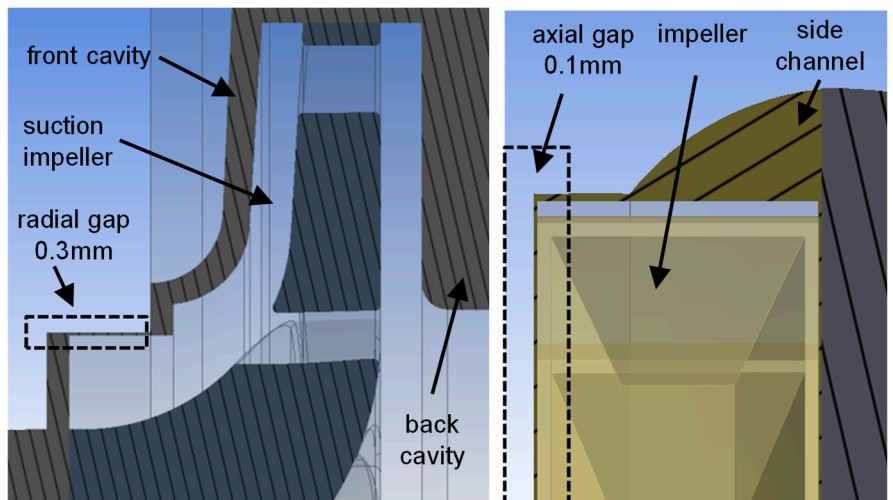

**Figure 4.** Narrow gaps at impellers.

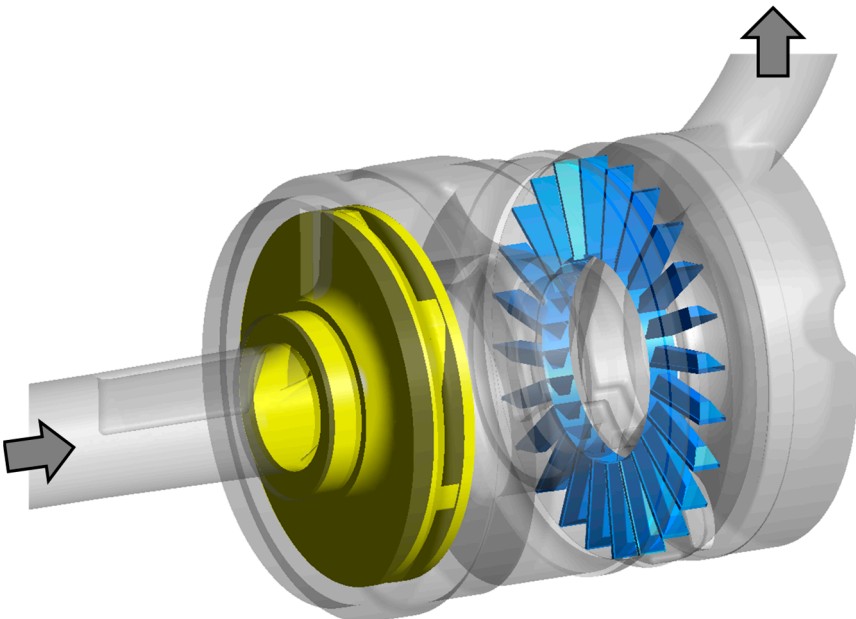

**Figure 5.** Full computational fluid dynamics (CFD) model of the analysed original side channel pump with radial suction runner (**left**) and impeller (**right**).

## 3.2. Numerical Model of the Side Channel Pump

Based on numerous successfully performed projects in the field of hydraulic fluid machinery at the institute within the last decade [15–18], a wealth of experience was gained. The simulations of the investigated pump are primary based on this knowledge, especially the definitions of the numerical model and the solver (turbulence modelling, boundary conditions, interfaces, iterations . . . ), and approaches of modelling hub and shroud cavities.

As stationary simulations mostly give a good first impression on the flow situation it is vital for getting the most accurate results, to also perform transient simulations. The turbulence model used for stationary simulations (Reynolds-Averaged Navier–Stokes, RANS) was k-$\omega$-SST-turbulence model (Shear Stress Transport turbulence model) from Menter [19], combining the 2-equations-turbulence-models k-$\varepsilon$- and k-$\omega$. This turbulence-model requires an adequate mesh for modelling near wall (k-$\omega$) and wall distant (k-$\varepsilon$) flow regimes.

Though the SST-turbulence model could be used for transient simulations (Unsteady Reynolds-Averaged Navier–Stokes, URANS), experience has shown that it does not always provide satisfying results, even if the grid and time step resolution would be sufficient for that purpose. That is why the scale resolving turbulence model SAS-SST(Scale-Adaptive Simulation Shear Stress Transport turbulence model) from Menter [20] was utilized. It combines the advantages of the SST-model in steady zones but enables to switch to SRS (Scale-Resolving Simulation) mode in flows with large and unstable separation zones which are likely to be present between impeller and side channel.

In stationary simulations, the interfaces between suction piece and impeller as well as impeller and side channel were modelled with the Frozen Rotor approach as simulations with the mixing plane approach were highly unstable due to the averaging which is just not applicable for the investigated zones. As the Frozen Rotor modelling assumes a fixed Rotor-Stator-position, five simulations with clocked positions—each with an impeller moved 3° further—were made and the final result represents the arithmetical average of all five simulations. During stationary simulations, between 600 and 2000 iterations were required depending on the convergence of each operating point. The evaluation of the convergence was based on two factors. First on the RMS-weighted residuals and second on the behavior of the defined monitor points (e.g., different defined pump heads). When residual were sufficiently small and the fluctuations of the monitor points negligible, the simulations were stopped.

For the calculation of the results, the arithmetic average of the last 100 iterations was used. (For information, one stationary simulation took 20 h on a High-Performance Workstation with 16 cores)

For transient simulations, which used the corresponding stationary result for initial values, the Transient-Rotor-Stator approach with an adaptive timestep was used. The first two revolutions used a timestep which equals for 2° as shown in Equation (5).

$$t_{trans,0\text{--}720°} = \frac{2 \cdot \pi}{1450} \cdot \frac{2}{360} \tag{5}$$

Those two revolutions were used as settling time for the simulation. Afterwards the timestep was reduced to get a resolution of 1° for the next two revolutions, whereas only the last one was used for calculating the results. This timestep allows for a RMS Courant Number of around 5 but as CFX uses an implicit solver it therefore does not have a numerical stability limit based on Courant Number.

$$t_{trans,720\text{--}1440°} = \frac{2 \cdot \pi}{1450} \cdot \frac{1}{360} \tag{6}$$

### 3.3. Post-Processing

A major advantage of simulations compared to measurements is the infinite possibilities of the analysis of the results. Beside conventional measurement data like head curve, efficiency and shaft power it is possible to calculate quantities and data such as

- Pump head from inlet to outlet
- Velocity components (cm, cu, ... ) of the suction impeller
- Flow interaction between impeller and side channel
- Flow situation of all components to detect possible flow separation and backflow zones
- Leakage flow in narrow gaps

Given that kind of information, it is possible to get deeper insights into the functionality and behaviour of the pump and its components. The whole post-processing was carried out with ANSYS CFX Post 17.1 and some quantities are described in detail below.

Head-curve was calculated according to EN ISO 9906 as described in Equation (7).

$$H_{9906} = z_2 - z_1 + \frac{p_2 - p_1}{\rho \cdot g} + \frac{u_2^2 - u_1^2}{2 \cdot g} \tag{7}$$

Whereas the data in the defined positions, each two diameter before the inlet respectively after the outlet, were calculated at corresponding planes (PlaneInlet and PlaneOutlet). The definition of the head with the so called CFX expression language (CEL) uses the area averaged pressure and velocities (pump discharge divided by plane areas) at those planes.

Efficiency is defined as the ratio of the hydraulic power (pump head and discharge) by shaft power as described in Equation (8):

$$\eta = \frac{\rho \cdot g \cdot Q \cdot H}{P_{shaft}} \tag{8}$$

The required shaft power is calculated on all wetted, rotating surfaces of the pump including hub and shroud cavities and gaps.

Pump head from inlet to outlet: needs defined positions to get an overview of the behaviour and the occurring losses inside. Those positions are

- Pump inlet
- Outlet suction impeller
- Inlet suction piece
- Impeller—0–360°

- Pump outlet

On all mentioned positions except the impeller, the total pressure on the stationary part of the interface was used to calculate the existing head. In the impeller, the following positions as shown in Figure 6 were defined and used for a deeper analysis of the head rise and interaction of impeller and side channel.

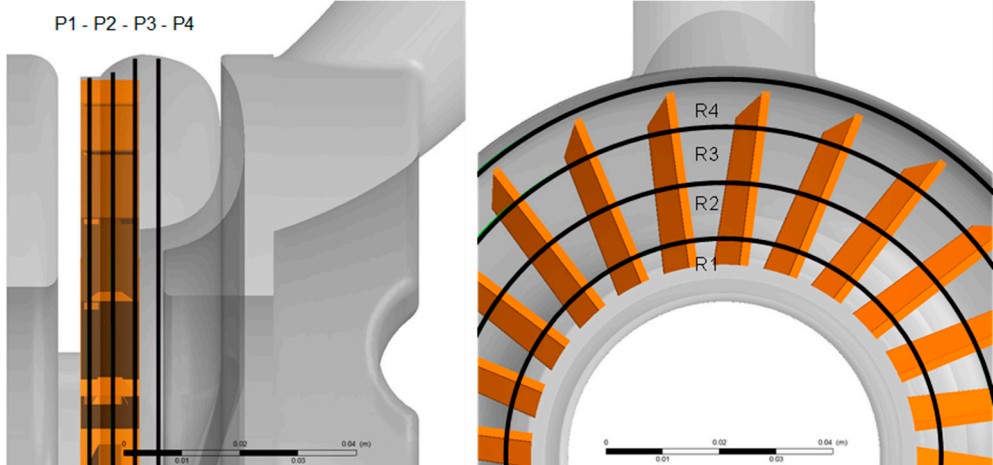

**Figure 6.** Planes (**left**) and radii (**right**) for calculation of total energy inside the impeller.

First, four different planes were placed across the impeller and side channel (Figure 6 left):

- Plane 1: 1 mm distance in axial direction from suction side
- Plane 2: in the middle of the impeller
- Plane 3: 1 mm distance in axial direction from pressure side
- Plane 4: in the middle of the side channel

Additionally the pressure/pump head was calculated on four different radii (Figure 6 right):

- Radius 1: $r = 40$ mm
- Radius 2: $r = 50$ mm
- Radius 3: $r = 60$ mm
- Radius 4: $r = 70$ mm (only for planes 3 & 4)

Given the data on all mentioned circles, it is possible to evaluate the pressure rise from inlet—which is defined as 0° position—to outlet—defined as 345° position. The missing 15° account for the interrupter which is required for the function of the pump itself, but is not of interest for the resulting pump head.

## 4. Results

The results of the pump head and efficiency are shown in Figure 7 for stationary and transient simulations. Though the pump head and efficiency near BEP show a satisfying correlation with the measurement data (measurements according to EN ISO 9906 Class 1) provided by the manufacturer [21], it can be seen that the transient results give a more accurate prediction of the real BEP position. Additionally, the transient head curve in part load is more reliable.

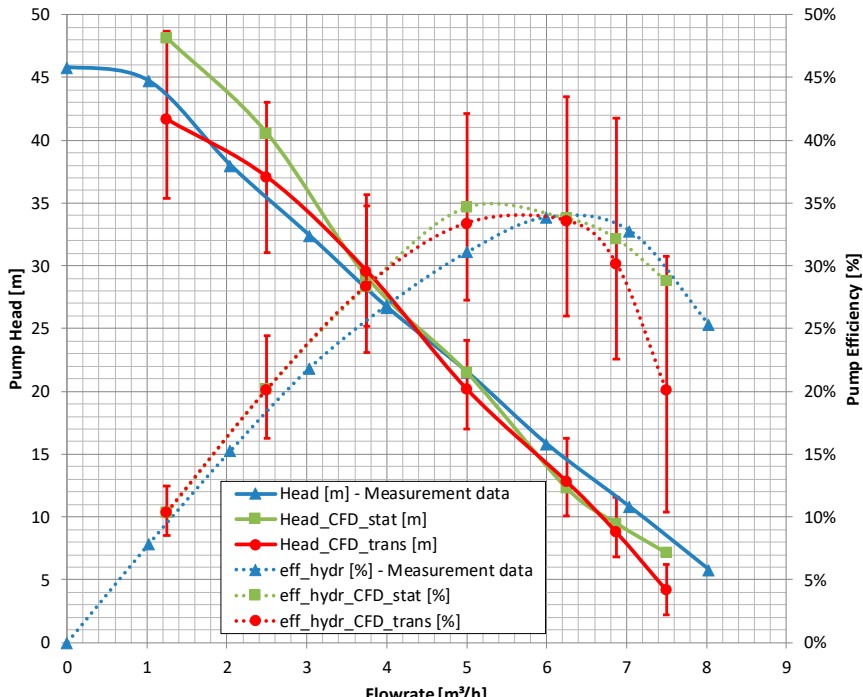

**Figure 7.** Comparison of measurement and computational fluid dynamics (CFD) results (stationary and transient).

In overload conditions it is surprising to see that stationary simulations are superior, in both head and efficiency. A possible explanation may be the occurring convergence behavior and the chosen, already described calculation which uses the average value of the last 100 iterations. As shown in Figure 8, the pump head heavily fluctuates in stationary mode—even in BEP. In transient mode the fluctuations can clearly be assigned to the impeller blades as shown in Figure 8, although the timestep needs to be sufficiently small to describe the real behavior.

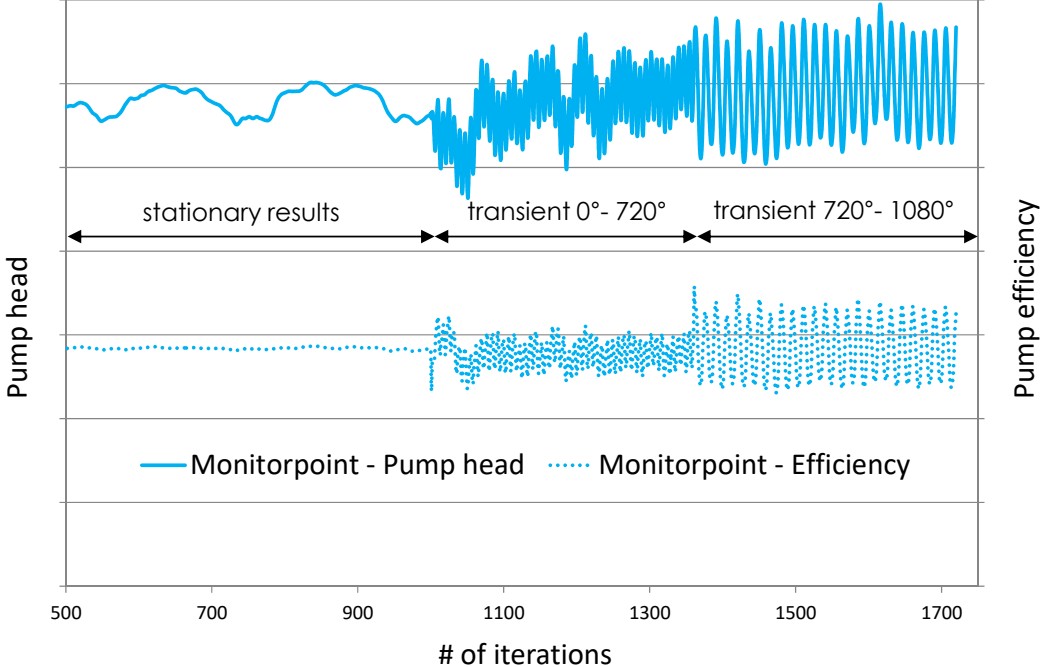

**Figure 8.** Monitor of head and efficiency at best efficiency point (BEP) for stationary and transient simulation.

In Figure 9, the total energy, calculated as an average value of all three planes at the four mentioned radii, is shown for BEP. Starting at the suction piece, the pressure only slightly increases within the first 60° (which represents four blade passages). In region 2, the significant energy rise of the pump is gained. Starting with a lower gradient at the beginning, a nearly linear behavior as theoretically predicted can be shown. When the transition from impeller to outlet begins (marked as zone 3 in Figure 9), the energy stays nearly constant until the interrupter.

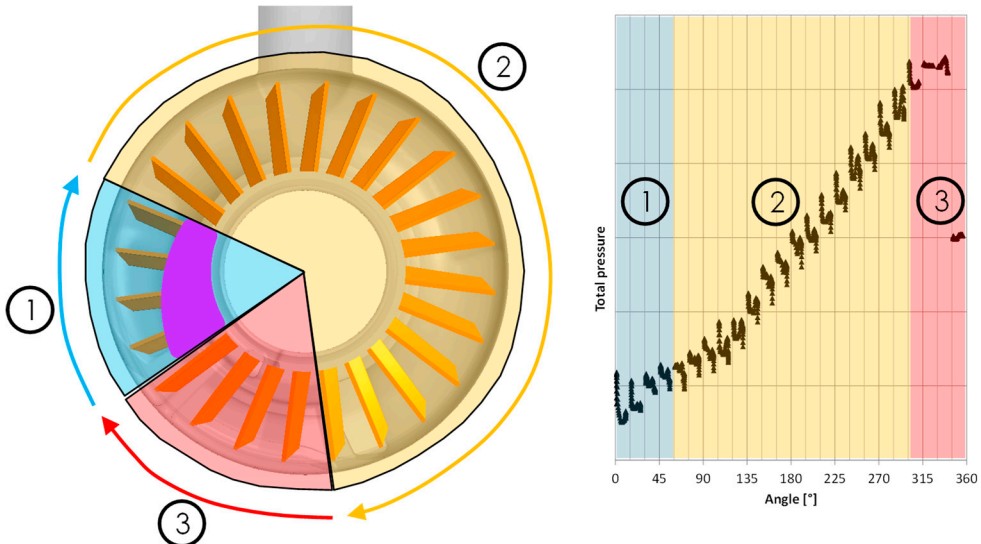

**Figure 9.** Total energy from impeller inlet to outlet at BEP.

A more detailed view of zone 2 at the linear characteristic is shown in Figure 10. At a constant radius, the total energy at all four planes is shown for six blade passages (each covers 15°). It can be seen that the highest pressure is reached near the small gap between impeller and suction side and then decreases towards the side channel. Inside the side channel, the pressure along a blade passage stays nearly constant and then increases across the passing blade.

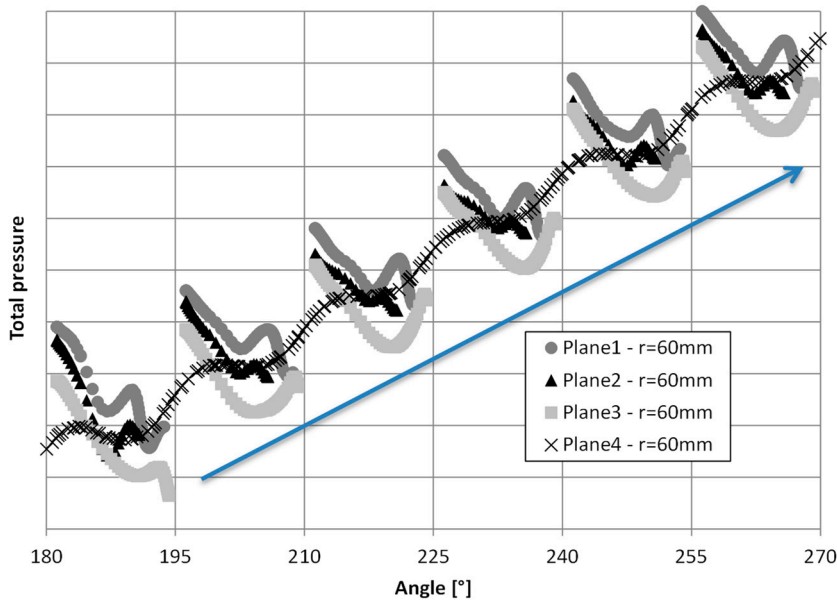

**Figure 10.** Distribution of total pressure at different planes—r = 60 mm—Q = 5 m$^3$/h.

Another outcome is the expected pressure decrease in the wake behind a passing blade. This phenomenon seems to be more distinct near the side channel as there is an additional pressure decrease due to significant inflow.

In Figure 11, the total pressure from pump inlet to outlet is shown for six operating points, with the pressure at inlet is defined at zero for all points.

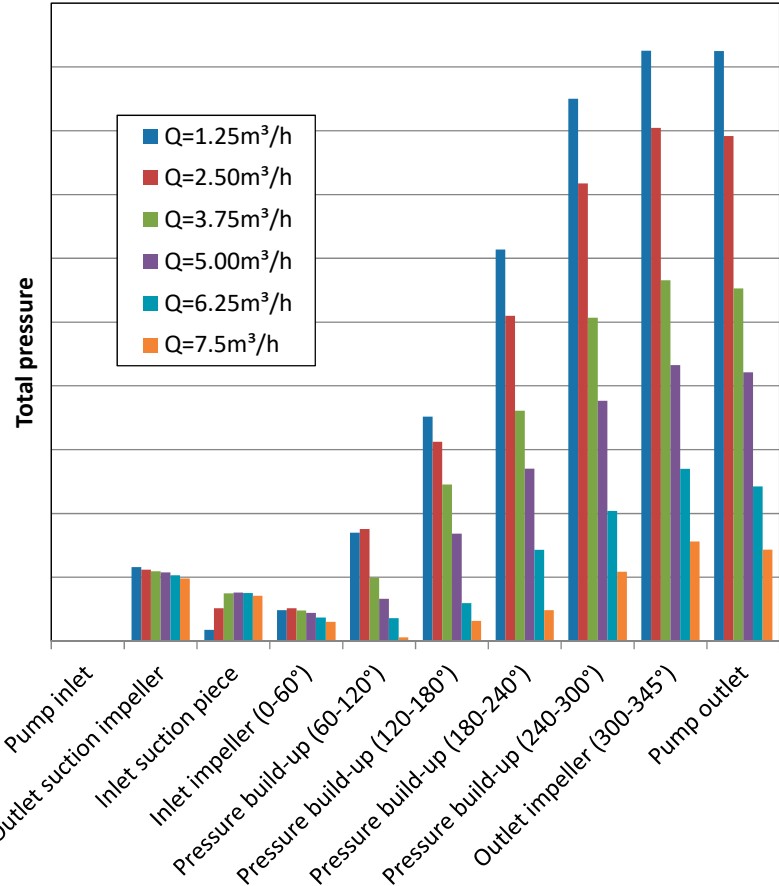

**Figure 11.** Total pressure from pump inlet to outlet at certain positions—stationary results.

At the inlet of the suction piece it gets more interesting. Though the suction stage delivers the highest head in deep part load of 1.25 m$^3$/h, the resulting head at the inlet of the suction piece is lowest. This yields from the energy dissipation inside the connecting passage between the outlet of the suction impeller and inlet of the suction piece as it consists of a quite big volume which is obviously not suitable for small flowrates.

Following the flow through the pump, the suction piece is the next part to be investigated. As shown in Figure 12, it can be shown that especially in part load a relatively large backflow exists (marked in blue color). Even around BEP a significant backflow occurs as shown in Figure 12b. Although this phenomenon cannot be eliminated entirely it should be minimized during the optimization process.

At the inlet region of the impeller between 0° and 60° the total pressure for all investigated operating points are almost identical. With increasing wrap angle the pressure rise shows the expected linear trend whereas the gradient gets steeper at decreasing flowrates.

Near the outlet of the impeller, the pressure rise declines and stays almost constant until the outlet of the pump.

This analysis gives a good insight in the optimization potential of the investigated pump which identifies the connecting passage between outlet of the suction impeller and inlet of the suction piece

as a main cause of losses. Additionally, the outlet of the impeller could be improved to get more head out of the last three blade passages.

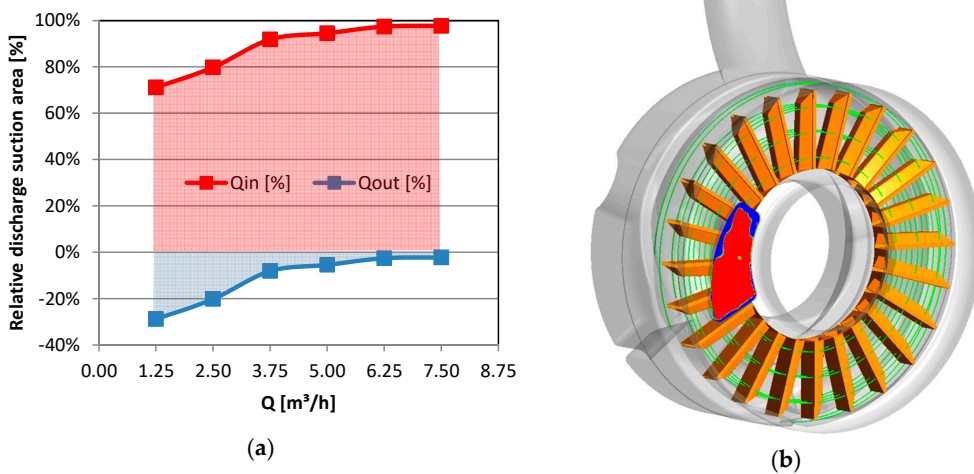

(**a**)                    (**b**)

**Figure 12.** (**a**) Relative discharge at suction area; (**b**) Inflow (red) and outflow (blue) areas at BEP.

## 5. Optimisation

Besides optimising the connecting passage between outlet of the suction impeller and inlet of the suction piece and the transition from impeller to outlet, it is vital to understand the behaviour and interaction of impeller and side channel. Due to the high numerical effort of the presented "full" numerical model of the side channel pump; this may not be suitable for optimization tasks. Therefore a simplified model is presented to analyse various geometrical variations of impeller and side channel which are then validated in the full model again.

### 5.1. Numerical Model for Optimisation

According to investigations from Fleder [3], a simplified numerical model was generated as shown in Figure 13. This model, which is completely parametric, consists of a straight inflow region with a cross-section area following the original one. The impeller and side channel were completely parametric and only the gap between impeller and suction side remained unchanged. The outlet region was modelled as a straight circular pipe. Given those simplifications it was possible to generate a reliable model (as proven with a mesh study) with only two mio. nodes.

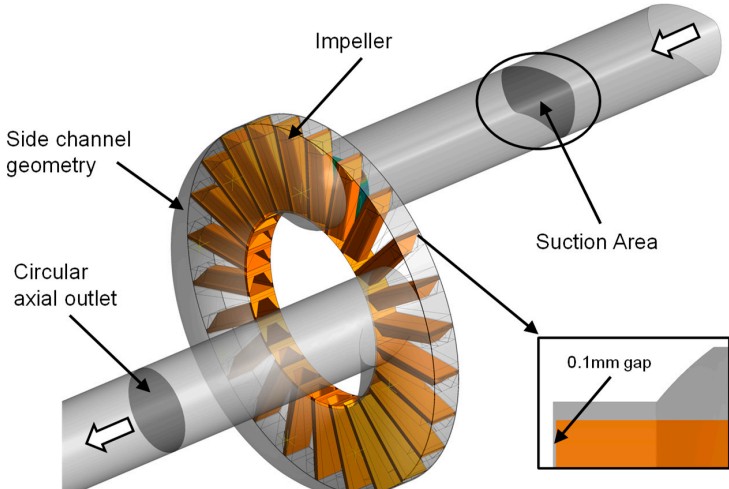

**Figure 13.** Simple model of side channel pump for optimisation.

The main parametrised components of the pump model are (in streamwise direction)

- Suction area: cross-section with four parameters as shown in Figure 14 right
- Impeller (simple): fixed diameters with five parameters as shown in Figure 14 left
- Side channel: two different side channel geometries with three resp. Six parameters as shown in Figure 15.

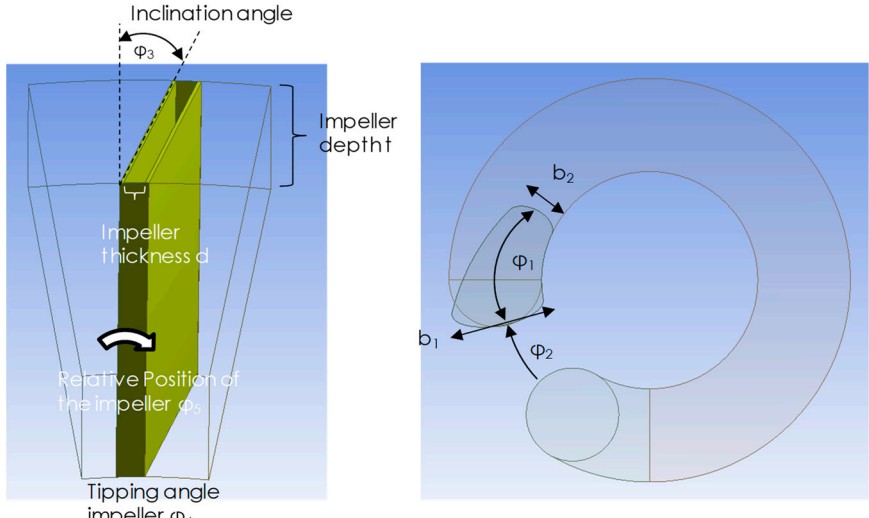

**Figure 14.** Parameters of impeller and suction area for optimisation.

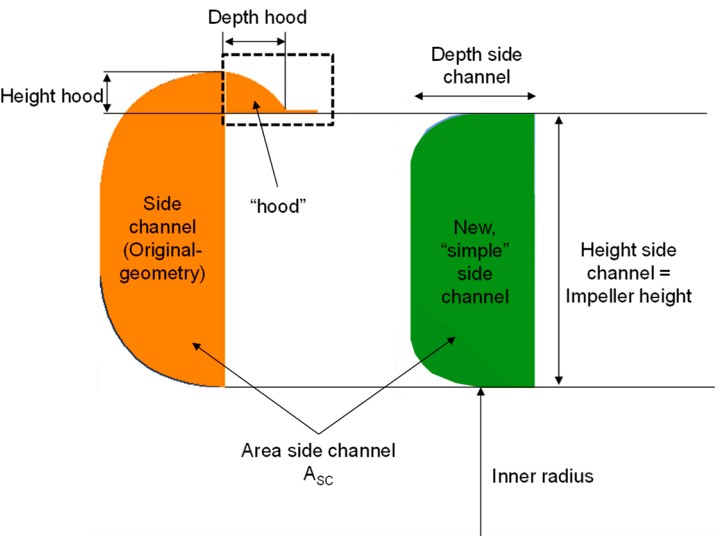

**Figure 15.** Parameters of side channel geometry for optimisation.

With the given parameters, more than 300 models were generated. For each model, at least three flow rates (BEP, 75% BEP, 125% BEP) were simulated. After an analysis of all $Q/H/\eta$ characteristics, for the 20 most interesting models the full characteristics with seven operating points were simulated.

*5.2. Results of the Optimisation Model*

As presenting all relevant results would go beyond the scope of this paper, only some selected examples are shown. In Figure 16, the effect of the impeller inclination angle is shown. As all other parameters were identical, the influence is impressive. Though the difference between the pictured angles is huge ($\pm 25°$) investigations of inclination angles in between fit perfectly in the shown trend.

For the given side channel and suction area, it seems that a straight impeller blade yields the optimum efficiency and highest head. However not only head and efficiency depend on the inclination angle, but also the position of the BEP.

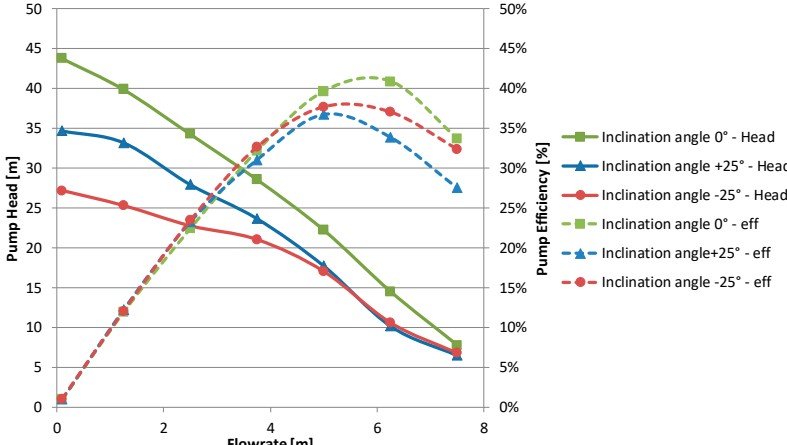

**Figure 16.** Effect of impeller inclination angle on pump head and efficiency.

This is interesting, because when varying the impeller depth and leaving all other parameters identical as shown in Figure 17, the position of the BEP remains unchanged although head curve and maximum efficiency vary.

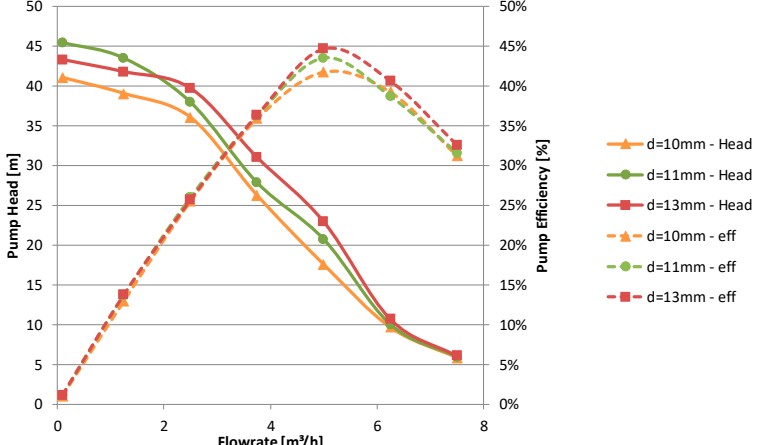

**Figure 17.** Effect of impeller depth on pump head and efficiency.

Another interesting fact is the identical efficiency up to BEP for all impeller variations though head curves differ significantly. This was already observed in Figure 16 which leads to the presumption that the impeller only has limited influence on efficiency (up to BEP) but strong effect on head curve.

The dimensions of the side channel have a major influence on head and efficiency as shown in Figure 18. Although the side channel area was only changed by ±10% a smaller one yields a higher maximum efficiency at a lower flowrate respectively a larger one a lower maximum efficiency at overload conditions. A smaller side channel also shows a steep efficiency decrease at overload conditions. Despite a similar head characteristic, the head around BEP heavily varies.

As described in chapter 2, up to now there are still two theories trying to describe the functionality of a side channel pump. If theory of angular momentum would be appropriate, pump head should be depending on blade (tipping) angle as shown in Figure 19. A positive tipping angle—the so-called forward curved blade—would yield a higher head than a backward curved one. Though this is not

common for radial pumps as blade loading increases and flow hardly follows the blade channel, its influence was investigated during the optimisation process as some literature suggests it [2,12,13].

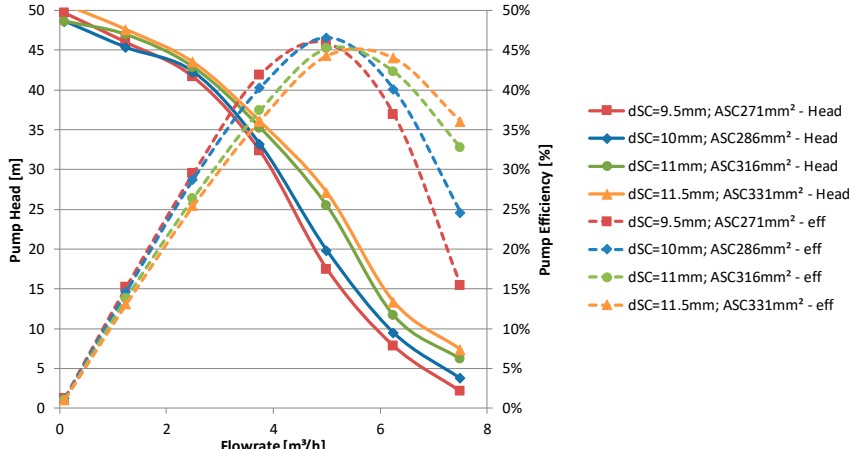

**Figure 18.** Effect of side channel depth and area on pump head and efficiency.

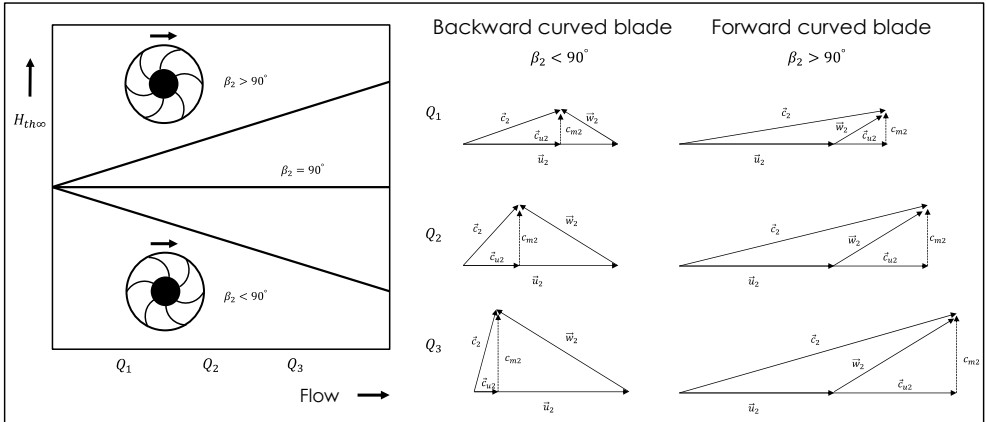

**Figure 19.** Theoretical pump head depending on blade tipping angle for a radial impeller.

It is shown in Figure 20 that for a certain configuration pump head stays almost constant for significantly varying blade tipping angles. This leads to the assumption that the theory of angular momentum is not entirely applicable for side channel pumps. Interestingly the pump efficiency and the location of the best efficiency point is clearly influenced by the blade tipping angle. A lower tipping angle yields a higher the efficiency at lower flowrates. This may be useful when trying to shift the location of the BEP.

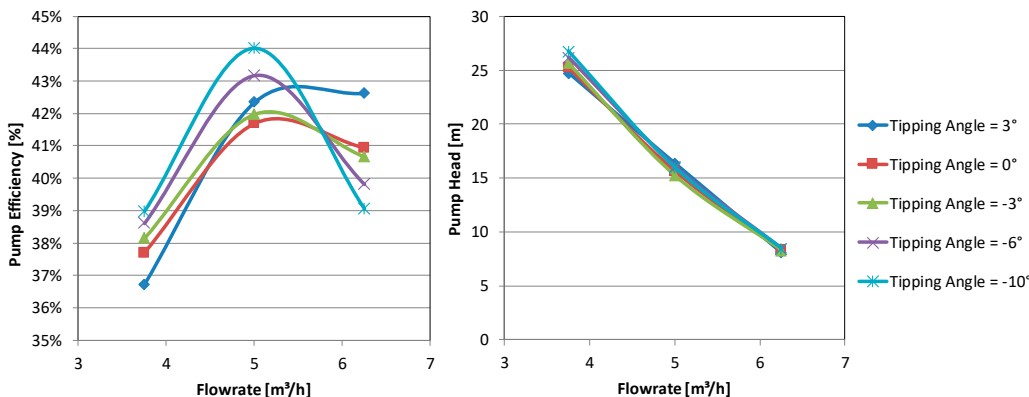

**Figure 20.** Effect of blade tipping angle on pump head.

As the pump may operate at different rotational speeds it is of major interest if there are certain phenomena or if pump head and efficiency can be estimated according to similarity laws for discharge as described in Equation (9) or head as described in Equation (10) like radial or axial pumps.

$$Q_{analytic,n2} = \frac{n_2}{n_1} \cdot Q_{n1} \tag{9}$$

$$H_{analytic,n2} = \left(\frac{n_2}{n_1}\right)^2 \cdot H_{n1} \tag{10}$$

A comparison of numerical and analytical calculation of pump head and efficiency for 1750 rpm, based on a rotational speed of 1450 rpm, is shown in Figure 21. It can be shown that at least for relatively small variations (+20%) of the rotational speed, the application of similarity laws is acceptable and reliable.

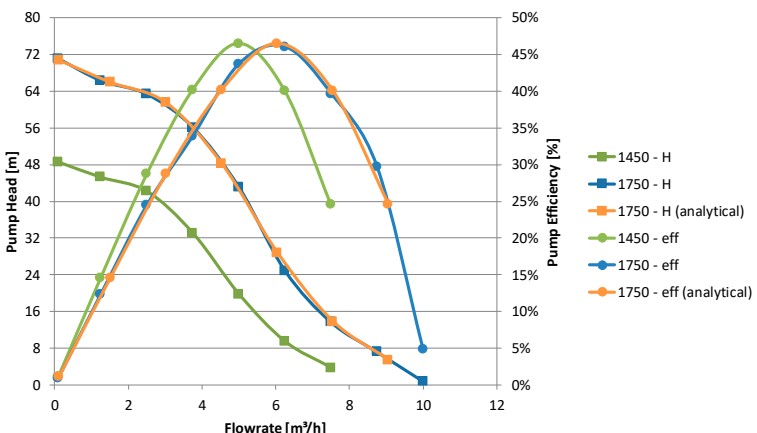

**Figure 21.** Pump head and efficiency for different rotational speed—numerical vs. analytical.

Finally a comparison of the results for the used simplified optimization model with the full model was carried out. As for the optimization only three operating points around BEP were investigated; it can be noted that those are quite in good accordance with the results of the full model as shown in Figure 22. In particular, the pump head in this region is predicted quite well. Due to the simplifications a quantitative match cannot be expected although the qualitative match—defined as the correct prediction of the location of BEP—is accomplished, which is of major importance for a reliable optimization.

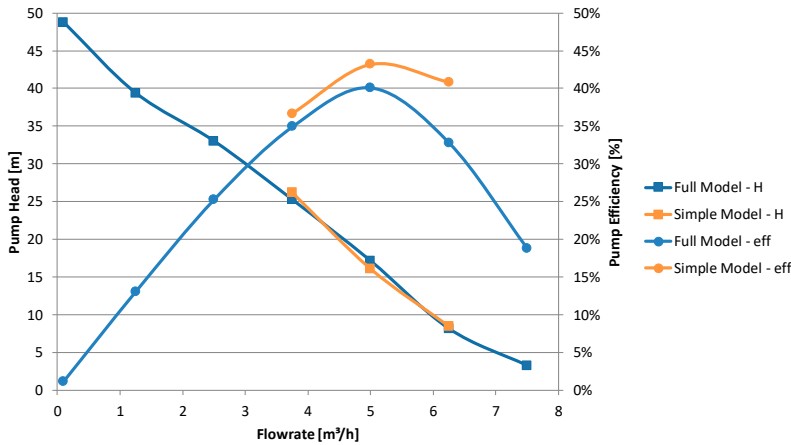

**Figure 22.** Resulting pump head and efficiency depending on model complexity.

*5.3. Optimisation Results of the Full Model and Comparison with Original*

The optimum geometry for achieving the identical head curve of the original pump and increasing the efficiency significantly was found in the simple model. Although the results of the simple model are quite reliable, it was vital to prove the performance in the full model, as the suction impeller and inflow and outflow conditions presumably influence the behavior of the pump. As not only the suction area, impeller, and side channel were changed but also the connecting passage between outlet of the suction impeller and inlet of the suction piece, simulations in the full model were necessary anyhow.

Figure 23 shows the impressive efficiency increase of the optimised pump whilst ensuring a nearly identical head characteristic at the same time. Besides increasing efficiency more than 30% (relatively) between deep part load and BEP, it is important to notice that the position of BEP was also shifted to slightly lower flowrates, as required from the manufacturer. It can also be seen that numerical simulations, both for the original and the optimised pump, predict the position of BEP at lower flowrates. The measurements also show a bit lower absolute value of the hydraulic efficiency which is probably due to neglected wall roughness of the rotating parts in the CFD-simulations.

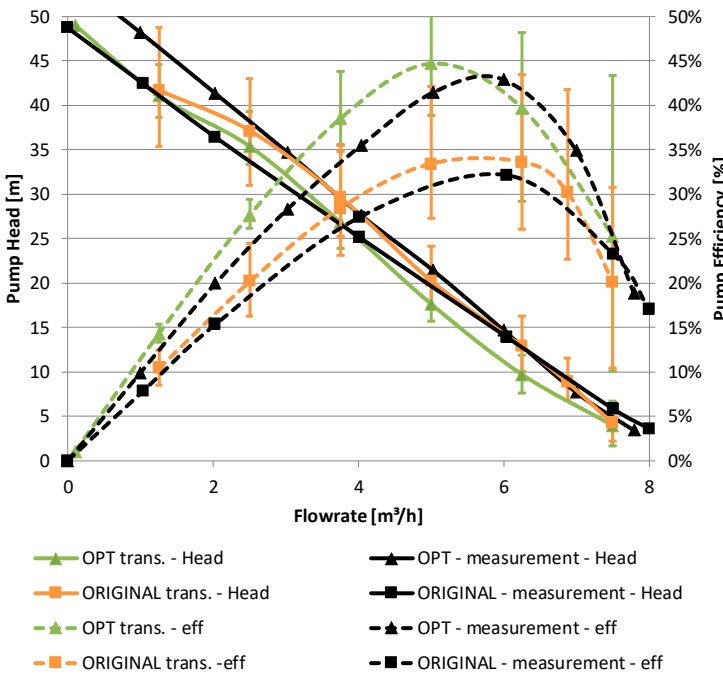

**Figure 23.** Comparison of original and optimised pump.

## 6. Conclusions

Though side channel pumps are rather known more as a mandatory powerful workhorse in industrial facilities than an efficiency talent, the presented paper shows how to get the most out of them. By using a large numerical model to cover all effects inside, it is possible to identify the reasons for the losses and eliminate them. As soon as the behaviour and the interaction of impeller and side channel are known, it is possible to adapt head curve, position of BEP, and improve the maximum efficiency. In parallel, all required gaps for safe operation and to comply with standards are already included.

In addition of optimising the presented pump, the paper also shows the strong influence of less known but important parameters like side channel depth, or less important assessed parameters like impeller depth, on head and efficiency.

**Author Contributions:** Conceptualization, M.M. and H.B.; Data curation, J.K.; Formal analysis, M.M.; Funding acquisition, H.B.; Investigation, M.M.; Methodology, M.M.; Project administration, H.B.; Resources, M.M.; Supervision, H.B. and H.J.; Validation, J.K.; Visualization, M.M.; Writing—original draft, M.M.; Writing—review & editing, H.B. and H.J.

**Funding:** This research received no external funding

**Acknowledgments:** We gratefully acknowledge the cooperation with Dickow Pumpen GmbH & Co. KG, who provided the pump, measurement data and many other information.

**Conflicts of Interest:** The authors declare no conflict of interest.

## Nomenclature

| | | |
|---|---|---|
| $A$ | [m$^2$] | area |
| $H$ | [m] | pump head |
| $Q$ | [m$^3$/s] | pump discharge |
| $P$ | [W] | power |
| $c$ | [m/s] | absolute velocity |
| $c_m$ | [m/s] | meridional velocity |
| $c_u$ | [m/s] | tangential part of absolute velocity |
| $g$ | [m/s$^2$] | gravitational constant |
| $n$ | [rpm] | rotational speed |
| $n_q$ | [rpm] | specific speed |
| $p$ | [Pa] | pressure |
| $r$ | [m] | radius |
| $t$ | [s] | timestep for transient simulations |
| $u$ | [m/s] | velocity |
| $z$ | [m] | geodetic head |
| $\delta$ | [-] | specific diameter |
| $\eta$ | [-] | efficiency |
| $\rho$ | [kg/m$^3$] | density |
| $\sigma$ | [-] | specific speed |
| $\varphi$ | [-] | flow number |
| $\psi$ | [-] | pressure number |
| $\omega$ | [1/s] | angular velocity |

### Subscripts and Superscripts

| | |
|---|---|
| 1 | pump inlet |
| 2 | pump outlet |
| 9906 | referring to DIN EN ISO 9906:2013-03 |
| o | at the outer radius of the impeller |
| i | at the inner radius of the impeller |
| SC | side channel |
| shaft | properties at pump shaft |
| trans | transient |

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
