# Peer review of "Maximum Efficiency Despite Lowest Specific Speed—Simulation and Optimisation of a Side Channel Pump†"

_ijtpp, doi:10.3390/ijtpp4020006_

Reviewer 1 Report

The paper is very good, near excellent, but let a few remarks be given below:

- all diagrams are without values on the axes, the numbers must exist or the diagrams are useless, esp. when the paper claims maximum efficiency and no value is given.

- inflow and outflow conditions for CFD analysis are not specified, although it is stated in line 394 that their influence is very significant

- in fig.23 legend is bw, but should be in color

- in Fig.22 H and eff in legend are vice-versa

Author Response

Thank you very much for your comments - I highly appreciate the effort you put in it!

-             all diagrams are without values on the axes, the numbers must exist or the diagrams are useless, esp. when the paper claims maximum efficiency and no value is given.

I understand your desire and it will be changed where possible (actually values were removed due to classification reasons). Although some diamgrams are presented to show the relative changes (Figures 8, 10, 11) and absolute values are not of importance.

-             inflow and outflow conditions for CFD analysis are not specified, although it is stated in line 394 that their influence is very significant

will be added

-             in fig.23 legend is bw, but should be in color

will be changed

-             in Fig.22 H and eff in legend are vice-versa

will be changed

Reviewer 2 Report

This reserach seems to be very intersting. However, it would be nice to explain and add following:

Values on the axis on Fig. 7. It nice to see the character, but also the velues.

Where did you obtain experimental results and following which international standard?

Frames around figures should be erased.

In page 15 you mentioned: "Interestingly the pump efficiency and the location of the best efficiency point is clearly influenced by the blade tipping angle. A lower tipping angle yields a higher the efficiency at lower flowrates.This may be useful when trying to shift the location of the BEP." Whar are those values of beta2 when you expect this ti happen? And in addition, what do you expect to happen for forward curve blades?

You also have written in page 15: "It is shown in Figure 20 that for a certain configuration pump head stays almost constant for significantly varying blade tipping angles. This leads to the assumption that the theory of angular momentum is not entirely applicable for side channel pumps". Do you suggest any modification of the Euler equation for turbomachinery, so it could be applied for the side channel pumps?

In the statement in the page 16 "Especially pump head in this region is predicted almost perfectly", word "perfectly" should be omitted.

Colors in legend on the figure 23 should correspond those in diagram.

Could you discuss flow phenomena which occur inside the pump?

General comment: It would be nice to see numbers in all diagrams. 

In Nomenclature: Meridional velocity should be noted with "m" in index. Similar for "u" in cu and "q" in nq.

References should be translated in English.

More references should be discussed in the introduction.

Author Response

Thank you very much for your comments - I highly appreciate the effort you put in it!

-             Values on the axis on Fig. 7. It nice to see the character, but also the velues.

will be added

-             Where did you obtain experimental results and following which international standard?

experiments were conducted at the laboratory of Dickow Pumpen and they were made according DIN 9906 - class 1 - added in the paper

-             Frames around figures should be erased.

will be changed

-             In page 15 you mentioned: "Interestingly the pump efficiency and the location of the best efficiency point is clearly influenced by the blade tipping angle. A lower tipping angle yields a higher the efficiency at lower flowrates. This may be useful when trying to shift the location of the BEP." What are those values of beta2 when you expect this to happen? And in addition, what do you expect to happen for forward curve blades?

informations added - tipping angle > 0° corresponds to a forward curved blade

-             You also have written in page 15: "It is shown in Figure 20 that for a certain configuration pump head stays almost constant for significantly varying blade tipping angles. This leads to the assumption that the theory of angular momentum is not entirely applicable for side channel pumps". Do you suggest any modification of the Euler equation for turbomachinery, so it could be applied for the side channel pumps?

Actually we are trying to adapt the existing design guidelines for side channel pumps to include influences like that. But this will take some more research and investigations and can therefore not be added in this paper.

-             In the statement in the page 16 "Especially pump head in this region is predicted almost perfectly", word "perfectly" should be omitted.

perfectly will be replaced by well/good

-             Colors in legend on the figure 23 should correspond those in diagram.

will be changed

-             Could you discuss flow phenomena which occur inside the pump?

I understand your interest, but as this paper has already 19 pages, this would require at least 1-2 more pages to present a "deeper" look inside

-             General comment: It would be nice to see numbers in all diagrams.

I understand your desire and it will be changed where possible (actually values were removed due to classification reasons)

-             In Nomenclature: Meridional velocity should be noted with "m" in index. Similar for "u" in cu and "q" in nq.

will be changed

-             References should be translated in English.

I do not see this necessary as it is easier to find them when referenced in original as some of them are even not available in english language

-             More references should be discussed in the introduction.

will be taken into account

Reviewer 3 Report

I this well written paper the authors present the conclusions of the hydraulic optimization of a side channel pump for designer and modeler audience. The paper delivers valuable technical information and novelty, the use of English is clear and the figures are of good quality, therefore I recommend this paper for publication.

The following minor considerations/changes could further improve paper quality:

line 68 – “kinetic energy is applied to the medium” – please revise;

line 125 – “but each of them with a 1:1 interface to maximize the resulting mesh quality and to overcome the need of a mesh interface.” – may be, the expressions “conformal interface” and “arbitrary interface” would be more adequate;

equations (5-6) – the Currant number should be presents along with the angular step size, since the mesh size is not specified exactly.

line 202 – The use of area weighted averages of enthalpy usually leads to inaccurate results violating the energy conservation. The use of “mass weighted” averages is recommended: the local enthalpy needs to be multiplied by the local mass flow rate, rather than the surface area.

line 249 – The steady state simulation model seems to be superior over the transient model only in regard to efficiency.

Figure 15 – The rounding radius of the side channel in the simplified model seems to be very different from that of the original design. Which side channel profile was used in the final laboratory testing of the optimum case?

Author Response

Thank you very much for your comments - I highly appreciate the effort you put in it!

-             line 68 – “kinetic energy is applied to the medium” – please revise;

will be changed

-             line 125 – “but each of them with a 1:1 interface to maximize the resulting mesh quality and to overcome the need of a mesh interface.” – may be, the expressions “conformal interface” and “arbitrary interface” would be more adequate;

will be changed

-             equations (5-6) – the Currant number should be presents along with the angular step size, since the mesh size is not specified exactly.

a reference to CFL number was added and also the note that CFX, which uses an explicit solver, is not that sensitive to CFL numbers

-             line 202 – The use of area weighted averages of enthalpy usually leads to inaccurate results violating the energy conservation. The use of “mass weighted” averages is recommended: the local enthalpy needs to be multiplied by the local mass flow rate, rather than the surface area.

the area averaging of pressure is very important as it considers backflow regions in contrast to a pure massflow averagingand therefore corresponds to the way the tests were conducted in the laboratory

-             line 249 – The steady state simulation model seems to be superior over the transient model only in regard to efficiency.

in overload condition both, head and efficiency, are predicted more accurately with stationary simulations whereas overall, transient is superior

-             Figure 15 – The rounding radius of the side channel in the simplified model seems to be very different from that of the original design. Which side channel profile was used in the final laboratory testing of the optimum case?

correct - the radius is different, but investigations have shown it has an negligible effect on the performance. The final optimization was conducted with a shape like the simple model.

Round  2

Reviewer 2 Report

File ijtpp-418100-Review 2.docx is attached.
